# Anemoside B4 Exerts Hypoglycemic Effect by Regulating the Expression of GLUT4 in HFD/STZ Rats

**DOI:** 10.3390/molecules28030968

**Published:** 2023-01-18

**Authors:** Qin Gong, Jilei Yin, Mulan Wang, Chengliang Zha, Dong Yu, Shilin Yang, Yulin Feng, Jun Li, Lijun Du

**Affiliations:** 1School of pharmacy, Jiangxi University of Chinese Medicine, Nanchang 330006, China; 2National Engineering Research Center for Manufacturing Technology of Solid Preparation, Nanchang 330006, China; 3Institute of Traditional Chinese Medicine, Jiangsu Union Technical Institute Lianyungang Branch, Lianyungang 222007, China; 4School of Life Sciences, Tsinghua University, Beijing 100084, China

**Keywords:** anemoside B4, hyperglycemia, diabetes, GLUT4, PI3K/AKT

## Abstract

Anemoside B4 (B4) is a saponin that is extracted from *Pulsatilla chinensis* (Bge.), and Regel exhibited anti-inflammatory, antioxidant, antiviral, and immunomodulatory activities. However, its hypoglycemic activity in diabetes mellitus has not been evaluated. Here, we explored the effect of B4 on hyperglycemia and studied its underlying mechanism of lowering blood glucose based on hyperglycemic rats in vivo and L6 skeletal muscle cells (L6) in vitro. The rats were fed a high-fat diet (HFD) for one month, combined with an intraperitoneal injection of 60 mg/kg streptozotocin (STZ) to construct the animal model, and the drug was administrated for two weeks. Blood glucose was detected and the proteins and mRNA were expressed. Our study showed that B4 significantly diminished fasting blood glucose (FBG) and improved glucose metabolism. In addition, B4 facilitated glucose utilization in L6 cells. B4 could enhance the expression of glucose transporter 4 (GLUT4) in rat skeletal muscle and L6 cells. Mechanistically, B4 elevated the inhibition of the phosphatidylinositol 3-kinase (PI3K)/protein kinase B (AKT) signaling pathways. Furthermore, we confirmed the effect of B4 on glucose uptake involved in the enhancement of GLUT4 expression in part due to PI3K/AKT signaling by using a small molecule inhibitor assay and constructing a GLUT4 promoter plasmid. Taken together, our study found that B4 ameliorates hyperglycemia through the PI3K/AKT pathway and promotes GLUT4 initiation, showing a new perspective of B4 as a potential agent against diabetes.

## 1. Introduction

Saponins are the main active components of *Pulsatilla chinensis* (Bge.) Regel (Pulsatilla). Anemoside B4, a triterpenoid saponin, is a monomeric natural product isolated from pulsatilla. The high content of B4 in the saponins has been used as a quality control marker for pulsatilla. The previous studies of B4 mainly focused on its composition identification, biotransformation, and tissue distribution [1,2,3,4]. The updated study in recent years reported that B4 has anti-inflammatory, immunomodulatory, antiapoptosis, and autophagy-enhancement effects [5,6]. Its outstanding anti-inflammatory effect was used to suppress the inflammatory reaction of acute and chronic kidney injury, chronic obstructive pulmonary disease (COPD), lung injury, and colitis diseases [7,8,9,10,11,12]. It also has an antiliver cancer effect by inhibiting PI3K/AKT/mTOR pathway and plays a protective role in liver injury by regulating the mTOR/p70S6K-mediated autophagy [13,14]. In addition, B4 could inhibit the proliferation of neointima by inhibiting the PI3K/AKT and P38MAPK signaling pathways, which has the potential to treat occlusive vascular disease [15]. A recent study reported that B4 reduced the healing time of diabetic ulcers by inhibiting inflammation and promoting angiogenesis and collagen matrix deposition [16]. However, the hypoglycemic effect of B4 on diabetes has not been reported. Our previous research indicated that B4 has a preventive and therapeutic effect on diabetic complications. B4 improved diabetic nephropathy by regulating autophagy and alleviated diabetic retinopathy in rats through its anti-inflammation and antiapoptosis properties. We speculated that if B4 had pharmacological activity in diabetic rats in the preliminary study, a significant upregulation of GLUT4 was observed. Therefore, the hypoglycemic effect and the mechanism related to GLUT4 of B4 in diabetic rats were explored in this study. 

Hyperglycemia is the main clinical feature of diabetes mellitus. Abnormally elevated blood glucose puts the organs in a state of high osmolar pressure, leading to complications such as diabetic nephropathy, retinopathy, and ulcerated feet [17,18]. In the middle stage of diabetes development, while compensatory hyperinsulinemia is insufficient to maintain normal glucose homeostasis, insulin-mediated glucose uptake is impaired, especially in muscle, and glucose tolerance is also significantly reduced. Skeletal muscle is the most important tissue for glucose uptake with approximately 80% of glucose uptake in the entire body occurring in skeletal muscle [19], indicating that improving the utilization of glucose in the muscle has positive significance for blood glucose stability.

Glucose transporter 4 (GLUT4), a transporter specifically expressed in muscle tissue, can translocate from the intracellular vesicles to the plasma membrane to mediate glucose uptake in response to insulin stimulation or noninsulin stimulation such as exercise and hypoxia. Impaired function and the content of GLUT4 can disrupt blood glucose homeostasis. Studies have shown that GLUT4 gene knockout in the skeletal muscles of mice significantly reduces basal glucose uptake, insulin, and exercise-stimulated glucose uptake. GLUT4 overexpression effectively reduces blood glucose. The basal blood glucose level of the model mice decreased and remained stable through implanting GLUT4 overexpression cells into engineered muscle structures in diabetic mice [20,21,22,23], implying that the regulation of GLUT4 levels was important in glucose transport in skeletal muscle and played a key role in stabilizing blood glucose. In order to explore the hypoglycemic effect and mechanism of B4, we focused on the transporter GLUT4 and observed the effects of B4 on glucose and GLUT4 in hyperglycemic rats and skeletal muscle cells of L6 in vivo and in vitro.

## 2. Results

### 2.1. B4 Improved Hyperglycemia and Glucose Tolerance in HFD/STZ-Induced Diabetes Rats 

First, we investigated the therapeutic effect of B4 on hyperglycemia rats. B4 was injected intraperitoneally continuously for two weeks. Before administration, the model rats exhibited a drastic elevation of FBG levels compared with normal rats. This elevation was significantly restricted by B4. After 10 days of administration (at day 24), the FBG levels of the rats in the model+B4 group were significantly lower than that of the model group (Figure 1b). The lowering effect was maintained until the end of administration (Figure 1a), whereas normal rats in the normal group maintained stable blood glucose levels during the experiment. HbA1c, a product of glucose combined with hemoglobin, can reflect the average blood glucose level in the past 2–3 months. HbA1c was significantly increased in model rats and B4 showed a decreasing trend (*p* = 0.072) (Figure 1c). At the end of the experiment period, an OGTT test was performed. All rats were given 2.0 g/kg glucose by gavage. Rats in the model group showed abnormal glucose tolerance and blood glucose rose sharply after 30 min administration, remaining at a high level over the next 120 min. The blood glucose levels in the model+B4 group were reduced quickly. The elevated blood glucose was also greatly suppressed by Dapagliflozin (Figure 1d). In addition, the AUC value of the model+B4 group was significantly decreased compared to the model group (Figure 1e). These results suggested that B4 reduced FBG levels and ameliorated the impaired glucose tolerance in hyperglycemia rats.

### 2.2. B4 Does Not Reverse Body Weight Loss, the Increase in Water Intake and Food Intake in Hyperglycemia Rats

In addition to elevated blood glucose, increased water intake and food intake are also common features of diabetic rats. Figure 2 showed that the body weight of rats in the model group declined from the beginning of treatment (Figure 2a), food intake began to increase on day 18 (Figure 2b), and water intake was at a high level before drug administration (Figure 2c). These manifestations were consistent with diabetes. However, there were no significant differences in body weight, food intake, and water intake between the model group and the B4-treated group during the two-week treatment period. 

### 2.3. B4 Increased GLUT4 Expression in Skeletal Muscle Relating to PI3K/AKT Pathways

GLUT4 is the most critical transporter that mediates glucose transport, and it is specifically expressed in skeletal muscle. The increased expression of GLUT4 for transport may promote the utilization of glucose in the body. PI3K/AKT is the upstream signaling pathway of GLUT4. Therefore, we examined the expression of GLUT4, PI3K, and AKT, and the phosphorylation of both PI3K and AKT in skeletal muscle using a western blot assay. The results showed that treatment with B4 enhanced GLUT4 expression remarkably, and at the same time, B4 also upregulated the expression of pPI3K/PI3K and pAKT/AKT (Figure 3a). To further confirm the enhancing effect of B4 on GLUT4, we analyzed the GLUT4 content indicated by fluorescence intensity in the skeletal muscle using an immunofluorescence assay. Muscle fibers were lumps of different shapes, DAPI-stained nuclei (blue) were located at the margins, and GLUT4 (red) was found throughout the cytoplasm. Figure 3b displayed that the fluorescence intensity (red) in the muscle of the model group was weaker than that of the normal group. B4 could heighten red fluorescence, and there was an obvious increase in GLUT4 expression that existed between the B4-treated group and the model group. These results suggested that B4 modulated blood glucose by increasing GLUT4 expression and may be related to the PI3K/AKT signaling pathway. 

### 2.4. B4 Increased Glucose Uptake and Enhanced GLUT4 Expression in L6 Cells 

The L6 cell is a common cell line used as skeletal muscle. L6 myoblast cells differentiate into myobute, which has the biological functions of skeletal muscle. As shown in Figure 4a, the appearance of myotubes and muscle fibers can be found, and the changes were more obvious from the results of coomassie brilliant blue staining. Myogenin, a transcription factor indicating their differentiation into myotubes, and desmin, a muscle-specific intermediate filament, were upregulated on day four and day seven of differentiation (Figure 4b), indicating L6 cells have differentiated into myotubes.

Firstly, the cell viability of L6 was tested using a CCK8 kit, and B4 with different concentrations (1024, 256, 64, 16, 4, and 1 μg/mL) were treated for 24 and 48 h, showing no effect on the viability of L6 myoblasts (Figure 4c), which proved B4 has no cytotoxic effects in the above concentration ranges.

Next, the effect of B4 on cell glucose uptake was studied by a glucose kit with the oxidase method. Compared with the control group, B4 (4 μg/mL) ascended the glucose consumption in the cell culture medium and increased the rate of glucose consumption after 24 h of treatment (Figure 5a,b), indicating that B4 promoted the utilization of glucose. In order to further prove its effectiveness, we also detected the intracellular glucose content, and the results were consistent with the previous results. B4 could increase the glucose content in cells (Figure 5c), which further illustrated the role of B4 in promoting glucose uptake. To elucidate whether the increased glucose uptake was due to B4 enhancing GLUT4 expression, we measured the expression of the GLUT4 gene and the protein in L6 cells. The results displayed that B4 (4 μg/mL) noticeably upregulated the gene expression compared to the control group (Figure 5d). In addition, it also enhanced the protein expression of GLUT4 in a time-dependent manner (Figure 5e), suggesting that B4 induced an increase in the GLUT4 gene and protein level, promoting glucose uptake in the skeletal muscle cells.

### 2.5. PI3K/Akt Pathway Was Involved in B4-Induced Increase in GLUT4 Expression

In order to verify GLUT4 and its correlated signaling after B4 administration, we investigated the PI3K/AKT signaling pathway related to GLUT4 on the enhancement of GLUT4 expression by B4. At the cellular level, we proved that B4 increased GLUT4 expression, and at the same time upregulated the phosphorylation of PI3K and AKT (Figure 6a). Then, Wortmannin (WM), a specific small molecule inhibitor of PI3K was employed to explore whether the effect of B4 on the increase of GLUT4 was regulated by the PI3K/AKT pathway. We compared cell glucose consumption and AKT activity in L6 cells treated with the inhibitors in the presence of B4, or both. Compared with the control groups, in the presence of WM, glucose consumption was significantly reduced, and in the presence of B4, glucose consumption increased. The glucose consumption in the WM+B4 groups was significantly lower than that of the B4 groups but still higher than that of the WM groups (Figure 6b). Similar observations were found in the AKT and GLUT4 protein expression (Figure 6c). The inhibitor WM of PI3K could attenuate the effect of B4 on increasing glucose consumption however, it failed to completely suppress it. It is suggested that B4 promotes glucose uptake, partly through the PI3K/Akt signaling pathway which may be involved in other regulatory pathways of glucose utilization in L6 cells.

### 2.6. B4 Directly Regulated GLUT4 by Promoting GLUT4 Gene Transcription 

The above results of the inhibitor proved that B4 indirectly regulates GLUT4 expression through the PI3K/AKT pathway to improve glucose uptake. In order to verify the effect of B4 on GLUT4 directly, we carried out the experiment on GLUT4 gene transcription. A recombinant plasmid of the GLUT4 promoter replacing the CMV promoter of the vector pEGFP-N1 was constructed, and the schematic diagram of construction is seen in Figure 7a. The construction process, such as amplification for GLUT4 promoter, pEGFP-N1 for digestion, digestion of recombinant plasmids, and gene sequencing and alignment, is shown in Figure 7b–e. The recombinant plasmid was tagged with a green fluorescent protein (GFP) so that the initial activity of GLUT4 could be determined by detecting the mRNA expression of GFP and the fluorescence intensity. After the B4 treatment, the intensity of green fluorescent (GFP) was stronger than that of the transfection group, and there was almost no fluorescence in the nontransfected plasmid group (Figure 7f). Consistent with the results of fluorescence intensity, B4 upregulated the mRNA expression of GFP, while the expression of GFP in the nontransfected group was very low (Figure 7g), suggesting that B4 enhanced GLUT4 initial activity and promoted gene expression. In addition, we observed the effect of B4 on vector pEGFP-N1 and found that B4 had no effect on the GFP expression of the vector pEGFP-N1 (Figure 7h), which means that the regulating effect of B4 on GLUT4 was excluded from the influence of the vector itself. The above results indicated that B4 had a direct regulation effect on GLUT4.

## 3. Materials and Methods

### 3.1. Animals and Cell Lines

Male SD rats, weighing 160–180 g, were obtained from Hunan SJA Laboratory Animal Co., Ltd. (SCXK (Xiang) 2021-0002) and were allowed to acclimatize to their surroundings for 1 wk. The animals were housed in temperature- and humidity-controlled rooms under a 12 h light/dark cycle in the Laboratory of Barrier Environment (Jiangxi Bencao-Tiangong Technology Co., Ltd. (SYXK (Gan) 2018-0002), and provided with unrestricted access to rodent chow and drinking water. All the experiments were performed according to the US National Institutes of Health guidelines for humane animal use (NIH Publications, No. 8023, revised 1978). All procedures described were reviewed and approved by the Institutional Animal Care & Use Committee of Jiangxi University of Traditional Chinese Medicine (TCM) and the Animal Welfare & Ethics Committee of the Jiangxi University of TCM (approval ID: JZLLSC20220811). The experimental procedure strictly followed the guidelines of the Experimental Animal Welfare and Ethics of China. 

L6 cells (rat skeletal muscle) and human embryonic kidney cells 293T were purchased from the National Collection of Authenticated Cell Cultures (Shanghai, China).

### 3.2. Chemical and Regents

Anemoside B4, Batch No. 20161107, purity > 98% using HPLC determination, was presented by Professor Yu-Lin Feng from the Phytochemical Department of our university. Dapagliflozin (DAPA) tablets were purchased from AstraZeneca Pharmaceuticals LP (USA). Streptozotocin (STZ), Batch No. M2082-05, purity > 99%, was purchased from AbMole BioScience (USA). Glucose anhydrous power, AR, was purchased from Sinopharm chemical reagent Co., Ltd. HbA1c and Glucose kits were purchased from Leadman BioChemical Company (Beijing, China). DMEM high glucose medium was purchased from Procell Life Science Technology.

### 3.3. Animal Model and Treatment

Hyperglycemic rats received a high-fat diet containing (*wt*/*wt*) 60% fat, 20% protein, and 20% carbohydrate for 4 weeks. After one week on a high-fat diet, a single intraperitoneal injection of 60 mg/kg b.w. STZ (dissolved in 0.05 M citrate buffer before use, the injection was completed within 20 min of preparation) was conducted in hyperglycemic rats. Normal rats (*n* = 6) were fed with a normal diet and injected with equal volumes of citrate buffer vehicle (5 mL/kg). On days 3 and 7 after injection, the FBG level of all animals was assessed in blood collected from the tail tip using a blood glucose meter (Accu-check, Roche), and the rats fasted for 6 h before the blood glucose test. According to the last blood glucose levels, STZ-injected rats whose FBG levels were higher than 13 mmol/L were regarded as successful hyperglycemic model rats and they were randomly divided into three groups: model group (*n* = 9), model+ B4 (2.5 mg/kg, *n* = 8) group, and model+DAPA (0.1 mg/kg, *n* = 8) group. B4 was injected intraperitoneally with a volume of 5 mL/kg, Dapagliflozin was administrated intragastrically in a volume of 20 mL/kg, and rats in the model group received an intraperitoneal injection of saline. All rats were administered their agent every day for 2 weeks. At the end of the experiment, the rats were sacrificed under anesthesia, blood was collected from the arteriovenous plexus of the eyeball, and the serum was separated by centrifugation at 3000 rpm for 10 min. Then, the organs were immediately isolated and snap-frozen in liquid nitrogen. Finally, these samples were stored at −80 °C for examination. The experimental procedure is shown in Figure 8.

### 3.4. Body Weight, Fasting Blood Glucose, Water, and Food Uptake

During the drug treatment period, body weight and FBG were measured weekly. For the FBG test, rats fasted for 6 h (8:00–14:00) with free access to water. Water intake was monitored by measuring the volume of water taken with a cylinder at a fixed time point every day. Food intake was recorded by weighing the feed every two days.

### 3.5. Oral Glucose Tolerance Test

In the last days of the experimental period, an oral glucose tolerance test was performed on the rats that fasted for 6 h [24,25]. After 2.0 g/kg glucose oral administration, blood samples were collected from the venous plexus of the eyeball, at 0, 30, 60, and 120 min, serum was isolated from the blood sample to measure the blood glucose using an automatic biochemical analyzer (HITACHI, Tokyo, Japan). The area under the curve (AUC) of the blood glucose concentration was calculated based on the data collected during the OGTT.

### 3.6. Gene Expression

The expression of mRNA was carried out using a real-time PCR assay according to the reference [26]. Total RNA was extracted from muscles using an RNA extraction kit (Biotech, China) and reverse-transcribed to cDNA using the Fastquant RT Kit (Yeasen Biotech, China) according to the manufacturer’s instructions. Real-time PCR for specific genes was performed on a 7500 Real Time PCR system (Applied Biosystems, Waltham, MA, USA) using an SYBR Green Master Mix kit (Yeasen Biotech, China) according to the manufacturer’s instructions. β-actin was used as an internal control. The primers GLUT4 (sense: CGGATGCTATGGGTCCCTAC, antisense: ACCATTTTGCCCCTCAGTCAT) and β-actin (sense: CTCTGTGTGGATTGGTGGCT, antisense: GCTCAGTAACAGTCCGCCT) were designed with GenBank NCBI (https://www.ncbi.nlm.nih.gov/) accessed on 20 May 2021, and manufactured by GenScript Biotech Company (Nanjing, China). 

### 3.7. Western Blot Analysis in L6 Cells and Muscle Tissues

Protein expression was analyzed using a western blot as previously described [27]. The muscle tissues were lysed with 2% SDS (contained protease inhibitor cocktail and phosphatase inhibitor cocktail), and the protein was separated by an 8% sodium dodecyl sulfate-polyacrylamide gel electrophoresis and transferred to an NC membrane. Membrane blockade with 5% skim milk sealed at room temperature for 2 h and then incubated overnight at 4 °C with diluted primary antibodies. Primary antibodies against pPI3K (Y607, rabbit polyclonal antibody, ab182651), PI3K (rabbit monoclonal antibody, ab191606), and pAKT (s473, rabbit monoclonal antibody, ab81283), were purchased from Abcam (Cambrige, Cambs, UK); antibodies against AKT (rabbit polyclonal antibody, 9272S) and GLUT4 (mouse monoclonal antibody, 2213S) were purchased from CST (Danvers, MA, USA). Secondary antibodies were the HRP labeled goat antimouse IgG-HRP (7076S) and goat antirabbit (7074S) IgG-HRP provided by CST (Danvers, MA, USA) were used to incubate the membrane at room temperature for 1.5 h, the ECL luminescent solution (Thermo, Waltham, MA, USA) was used to visualize targeted protein bands and imaged in the imaging system (Bio-Rad ChemiDocXRS+, Hercules, CA, USA). Image J software was used to quantify the gray level. HSP90 (mouse monoclonal antibody, TA500494) was purchased from Origene (Rockville, MD, USA) and was used as an internal control. 

### 3.8. Tissue Immunofluorescence Analysis

Paraffin slides with a thickness of 5 μm were deparaffinized and rehydrated, then immersed in a citrate buffer (PH = 6) and heated at 100 °C for 5 min for antigen retrieval. After cooling, the slides were washed with PBS 1× and blocked with 5% normal goat serum (Solarbio, Beijing, China) for 2 h at RT. Then, the slides were incubated with a mouse monoclonal anti-GLUT4 (sc53566, Santa Cruz, Dallas, TX, USA) solution diluted in blocking solution overnight at 4 °C. The following day, the slides were washed with PBS (5 min × 3 times) and incubated with fluorescent-labeled secondary antibodies (Alexa Fluor 594 goat antimouse IgG (HUABio, Hangzhou, China) in the dark for 1 h at RT and later washed with PBS (5 min × 3 times). Next, the slides were covered using an antifade mounting medium containing DAPI with coverslips. The images of sections were taken using a confocal microscope (SP8, Leica, Wetzlar, HE, Germany).

### 3.9. Cell Culture, Differentiation, and Identification

L6 cells were cultured in DMEM high glucose medium (4.5 g/L glucose) supplemented with 10% fetal bovine serum (FBS) and 1% antibiotics. The passage was required when cells were grown to 80% confluence. Prior to the experience, myoblast cells were differentiated into myotubes. The cell was cultured in a DMEM medium containing 2% FBS for 6 days, and the medium was replaced every two days during this period. The differentiated L6 cell was used in a subsequent trial. All cells were incubated in a humidified incubator at 37 °C with ambient oxygen and 5% CO_2_.

Coomassie brilliant blue staining [28] was used to identify cell differentiation. On days 0, 4, and 7 of differentiation, cells were rinsed with PBS, fixed with 4% paraformaldehyde for 30 min, and washed with PBS. Followed by permeabilization with 1% Triton X-100 for 10 min and rinsing with distilled water. Then, stained with 2% coomassie brilliant blue solution for 15 min and washed with distilled water. Washing time was 5 min each time for 3 times. Finally, images were taken using an optical microscope (Leica, Germany).

### 3.10. Cell Viability

Cell viability was tested using a Cell Counting kit (CCK8, MCE, Shanghai, China). L6 cells were cultured in a 96-well plate. Cells were treated with PBS as the control group and treated with B4 (1024, 256, 64, 16, 4, 1 μg/mL) as the treatment group. Each group was repeated in parallel five wells. After 24 h and 48 h of treatment, the culture medium was discarded. Then, 100 μL CCK8 working solution (CCK8 reagent was mixed with culture medium without FBS at a ratio of 1:9) was added to each well and incubated at 37 °C for 1 h. Optical density (OD) was finally read at 450 nm using a microplate reader (Molecular Devices, San Jose, CA, USA). Cell viability (%) was calculated according to the following formula: OD of treatment group/ control group × 100%. Tree-independent experiments were conducted.

### 3.11. Glucose Uptake Assay

L6 cells were seeded in a 96-well transparent plate with low density in 100 µL DMEM. After 6 days of differentiation, as described above. L6 myotubes cells were treated with B4 (64, 16, 4 μg/mL), and PBS buffer dissolved in 100 μL DMEM medium, six wells were repeated in each group. After 24 h of treatment, the cell supernatant was taken, and the glucose concentration was detected by the glucose oxidase method using an automatic biochemical analyzer (HITACHI, Japan). Glucose consumption in the culture medium reflects intracellular glucose uptake. 

L6 cells were cultured in a 6-well transparent plate. There were four wells in the normal group and the B4-treated groups. After B4 (64, 16, 4 μg/mL) treatment for 24 h, cells were collected and the intracellular glucose was lysed according to the instructions of the glucose content assay kit (APPLYGEN, Beijing, China), the supernatant of lysate was used to measure the glucose content using the oxidase method, and the protein in the precipitate was detected by a BCA kit (Beyotime, Beijing, China). The amount of glucose per gram of protein was calculated and deemed as glucose uptake. 

### 3.12. PI3K Inhibitor

As a PI3K inhibitor, 100 nM wortmannin (MCE) was pretreated to L6 cells for 30 min and cells were treated with or without B4 (4 μg/mL) or insulin (100 nM) for 24 h. Cells were cultured in a 96-well plate for glucose uptake assay and cultured in a 6-well plate for western blot analysis.

### 3.13. Construction of GLUT4 Promoter Plasmid 

Referring to reference [29], the GLUT4 promoter recombinant plasmid was constructed using pEGFP-N1 plasmid as the vector, the CMV promoter of pEGFP-N1 plasmid was replaced by the GLUT4 promoter using double enzyme digestion (see Figure 7a). Rat GLUT4 (Slc2a4, NM_012751) sequences of its gene starting area (−2300–100bp) were extracted from the EPD website (https://epd.epfl.ch//index.php) accessed on 28 September 2021. Primers were designed for the GLUT4 promoter region with Primer Premier 5 software. Restriction sites and protective bases of endonuclides AseI and XholI were added to the 5 ends of the primers. So, sense primer GCCGGCGCATTAATTGGTTTGGAGGATGAATAG and antisense primer CCGCTCGAGCTGGGATTTCAAAGTGGG were synthesized by GenScript Biotech (Nanjing, China). First, gDNA was extracted from rat liver using a gDNA extraction kit (Beyotime Bio, Shanghai, China) according to the instructions, and used a template for PCP amplification (PCR program: 94 °C for 2 min, 35 cycles of 95 °C for 25 s, 60 °C for 25 s, 68 °C for 50 s, and 68 °C for 5 min). The PCR product was mixed with DNA loading buffer and subjected to 1% agarose gel electrophoresis in TAE buffer, as the sample migrated to the bottom of the gel, it was imaged under ultraviolet light on a chemical imaging system (Bio-Rad, USA). The gel with the target gene was separated for DNA fragment recovery using a gel purification kit (TIANGEN Biotech, Beijing, China). Then, purified products and pEGFP-N1 plasmid were digested respectively by AseI and XholI enzyme (NEB Bio, Ipswich, MA, USA ) in a 37 °C water bath for 1 h, followed by electrophoresis and purification. The digested product and pEGFP-N1 were ligated by using the T4 ligation enzyme and buffer (NEB Bio, USA), and incubated at 25 °C for 30 min. Next, the ligated product was transformed into *E.coli* Trans5α chemically competent cell (TransGen, Beijing, China) by the heat shock method according to the manufacturer’s protocol. Transformed *E. coli* were cultivated at 37 °C overnight on LB solid medium supplemented with Kanamycin (30 μg/mL). An individual clone was selected and inoculated into an LB liquid medium containing Kanamycin for proliferation in a 37 °C incubator with 180 rpm. Plasmid extraction was carried out by Plasmid Midi Kit (Omega, Norcross, GA, USA) and DNA concentration was determined by Nano Drop 2000 (Thermo, USA). Finally, the cloned plasmid was analyzed by restriction enzyme digestion and gel electrophoresis, and the cloning plasmid was sequenced (Shenggong Bio, Shanghai, China) to confirm the validity of the clone, which was named GLUT4 promoter recombinant plasmid.

The cloned plasmid was transfected into 293T with a Lipo3000 kit (Yeasen, Shanghai, China) referring to instruction. The experimental group was divided into transfected with GLUT4 promoter plasmid, pEGFP-N1 plasmid, and nontransfection groups. The transfection groups were treated or not treated with B4 (4 μg/mL) for 24 h and 48 h. GFP mRNA expression was examined using a qQCR assay to evaluate whether B4 promotes the activity of the GLUT4 promoter. 

### 3.14. Statistical Analysis

All values were expressed as mean ± S.E.M. Data were statistically analyzed using one-way analysis of variance (ANOVA) with F value determination. The F test was performed using GraphPad Prism 8.01 software (Graphpad, San Diego, CA, USA). Student’s *t*-test between two groups was performed after the F test. A P value of less than 0.05 was considered statistically significant. The statistical graph was produced using GraphPad Prism 8.01 software as mentioned above.

## 4. Discussion

In this study, we investigated the potential hypoglycemic effect of B4 in hyperglycemic rats induced by HFD combined with STZ. The results showed that B4 significantly reduced the blood glucose level and accelerated glucose metabolism in vivo, which was related to the increased expression of GLUT4. We demonstrated that B4 could enhance GLUT4 expression by activating the PI3K/AKT signaling pathway and promoting GLUT4 gene transcription. Our results provided experimental evidence for B4 prevention and the treatment of diabetes, as well as a molecular basis for elucidating the protective role of B4 in diabetes.

B4 is a pentacyclic triterpenoid saponin compound with 23-hydroxybetulinic acid as the mother nucleus structure, which was isolated from Pulsatilla. Our previous pharmacokinetic results showed that the bioavailability of oral administration of B4 was very low (about 0.2%), so the administration route of intraperitoneal injection was adopted in this study. In addition, the results of the hemolytic and anaphylactic studies found that B4 does not cause hemolysis and allergic reactions and there is a high safety for injection administration. Dapaglifozin, an SGLT2 inhibitor is a new drug used in the treatment of type 2 diabetes mellitus clinically. It has a significant hypoglycemic effect and is not affected by the hypofunction of pancreatic β cells and the degree of IR. Diabetes mellitus is a heterogeneous clinical entity, characterized by high blood glucose levels or hyperglycaemia, accompanied by “polydipsia”, “polyphagia” and “emaciation” symptoms [30,31]. Low glucose tolerance is a necessary stage in the development of diabetes and is also regarded as the gold standard for the diagnosis of diabetes [32]. Our results showed that in the hyperglycemia rat model caused by continuous high-fat diet and single STZ stimulation, blood glucose increased sharply, water intake and food intake were significantly increased, and weight loss appeared, which were consistent with the clinical symptoms of diabetes. B4 distinctly reduced the blood glucose of hyperglycemic rats, comparable to the effect of dapagliflozin. OGTT is the blood glucose level measured within 120 min after oral administration of a large amount of glucose, which can reflect the metabolism of glucose in vivo [33]. In the OGTT test, the blood glucose level of the model rats elevated observably at 30, 60, and 120 min after the glucose challenge. The AUC value also increased significantly, indicating that glucose tolerance was impaired in hyperglycemic rats. On the contrary, B4 inhibited the elevated blood glucose at each time point and reduced the AUC value, suggesting that B4 could promote glucose metabolism, improve impaired glucose tolerance, and lower blood glucose in hyperglycemic rats. In addition, B4 also tended to decrease HbA1c, an irreversible product of the combination of glucose and hemoglobin, which reflects the fasting blood glucose level within 2–3 months rather than transiently and is used to evaluate the control degree of diabetes. There was only a downward trend in the effect of B4 on HbA1c without significant diminishment, which may be due to the administration period of B4 not being long enough. These results confirmed the hypoglycemic effect of B4. Abnormally elevated blood sugar can lead to increased osmotic pressure in the kidneys leading to polyuria and polydipsia. In addition to sugar metabolism disorders, protein and fat metabolism disorders occur, resulting in weight loss. However, B4 did not reverse weight loss or increase water intake and food intake in hyperglycemic rats, which may be related to the effects of B4 on renal function and lipid metabolism. These aspects were not investigated in this study and need further indepth study. In addition, B4 had no effect on the food intake of model rats, insinuating that the lowering of blood glucose by B4 was not due to the influence of the animal’s food intake.

When blood glucose is elevated, the body secretes insulin to stimulate its peripheral sensitive tissues—muscles to take up glucose so as to increase the glucose going to maintain blood glucose stability. Skeletal muscle is the main tissue for glucose disposal, and it is responsible for 70–90% of glucose uptake in the blood in the postprandial state [34]. Glucose utilization in skeletal muscle is mediated by GLUT4, also known as solute transporter protein family 2 (SLC2A4), which is the most abundantly expressed and insulin-sensitive glucose transporter isoform in skeletal muscle and adipose tissue [35,36]. It plays a central role in facilitating glucose uptake in skeletal muscle to maintain blood glucose homeostasis [37]. In normal conditions, 20% of GLUT4 resides on the cell surface and 80% is stored in intracellular vesicles. When stimulated by insulin or muscle contraction, GLUT4 is translocated from the intracellular storage vesicles to the muscle surface, mediating glucose transport into the skeletal muscle [38,39]. Overexpression of GLUT4 significantly accelerated glucose transport in transgenic rats with or without insulin stimulation [40,41,42] and muscle glucose utilization was also increased, which effectively improved impaired glucose tolerance and reduced blood glucose levels in diabetic rats [23,43]. Therefore, promoting GLUT4 transport or expression to facilitate muscle glucose uptake plays a key role in lowering blood glucose. Our results showed that B4 upregulated the expression of GLUT4 protein and mRNA in the skeletal muscle of hyperglycemic rats. Similarly, B4 increased GLUT4 expression in L6 skeletal muscle cells, as well as promoted glucose consumption and increased glucose uptake. These data suggested that the hypoglycemic effect of B4 was closely related to the enhancement of GLUT4 expression.

The PI3K/AKT pathway is the upstream signaling pathway of GLUT4 and is dependent on insulin stimulation. Insulin specifically binds to the insulin receptor (InsR) on target cells to generate a series of signal transduction, which in turn activates insulin receptor substrate (IRS), PI3K, and AKT signal molecules. The activation of AKT phosphorylates AKT substrate of 160 kDa (AS160), which acts on GLUT4 to enable tissue to utilize glucose [44,45]. Both in vitro and in vivo results showed that B4 could promote PI3K/AKT phosphorylation. To further explore whether the enhancement of B4 on GLUT4 was indirectly regulated by PI3K/AKT or had a direct effect on GLUT4 transcriptional activity, we used Wortmannin, an inhibitor of the PI3K, and constructed a recombinant plasmid of the GLUT4 promoter to elucidate the pathway by which B4 acts on GLUT4. We found that B4 increased the expression of GLUT4 and promoted glucose uptake in addition to the regulation of the PI3K/AKT pathway. Interestingly, B4 also enhanced the activation activity of GLUT4 and promoted gene transcription, suggesting that B4 may directly affect the promoter region of GLUT4. These results implied that the effect of B4 on GLUT4 was multipathway and the hypoglycemic mechanism of B4 was complex. What is interesting is whether B4 directly binds to the promoter of GLUT4 or acts on transcription factors binding to the site on the GLUT4 promoter, thereby initiating GLUT4 expression, which is worthy of further study. 

## 5. Conclusions

To sum up, B4 promotes the gene transcription of GLUT4 and regulates the PI3K/AKT signal pathway, enhancing the expression of GLUT4 which contributes to the facilitation utilization of glucose in muscle cells, thus improving hyperglycemia. This study provides an experimental basis for B4 against diabetes, which is a new pharmacological activity of B4.

## Figures and Tables

**Figure 1 molecules-28-00968-f001:**
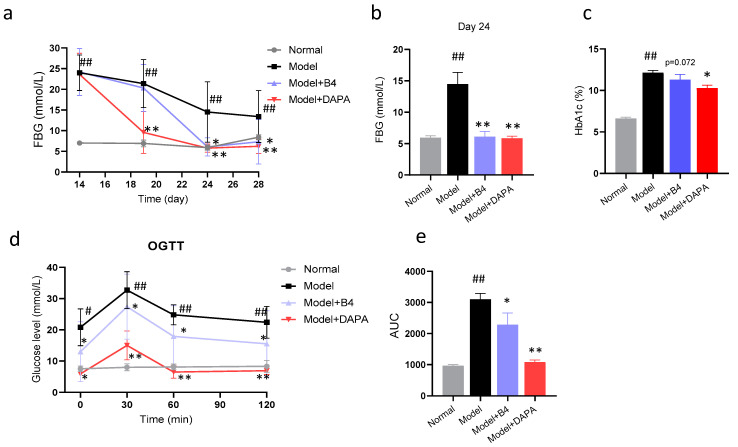
Effect of anemaside B4 (B4) on blood glucose of hyperglycemia rats. (**a**) FBG level on different days after administration; (**b**) FBG level after 10 days administration; (**c**) HbA1c level at the end of administration; (**d**) FBG levels at different times in the oral glucose tolerance test (OGTT); (**e**) The AUC of OGTT. # *p* < 0.05, ## *p* < 0.01 vs. Normal group; * *p* < 0.05, ** *p* < 0.01 vs. Model group.

**Figure 2 molecules-28-00968-f002:**
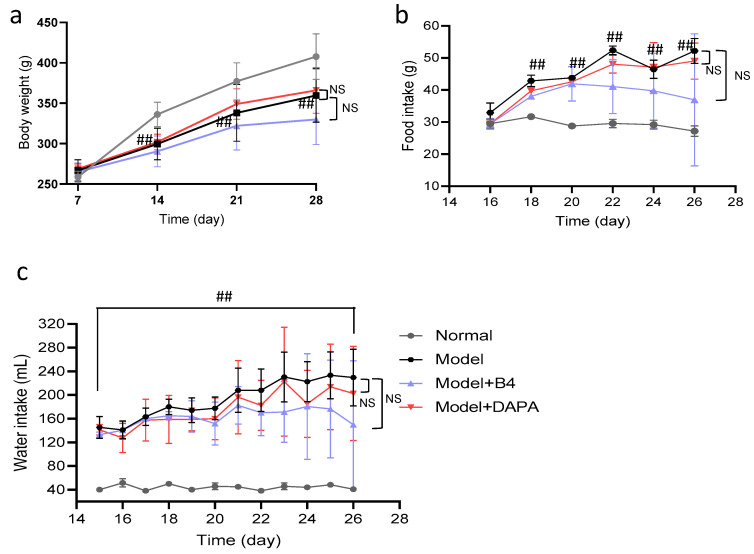
Effect of AB4 on body weight, food intake, and water intake of hyperglycemia rats. (**a**) body weight; (**b**) food intake; (**c**) water intake. ## *p* < 0.01 vs. normal group. NS: no significance.

**Figure 3 molecules-28-00968-f003:**
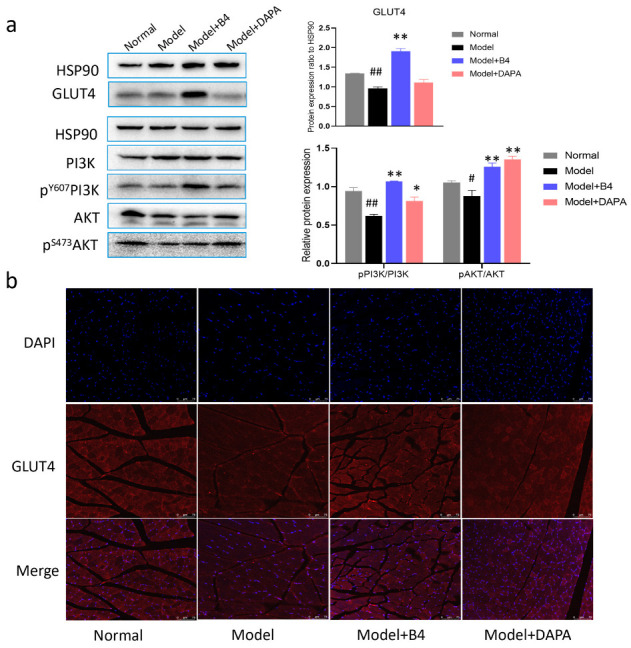
The expression of GLUT4 and phosphorylation of PI3K and AKT in skeletal muscle. (**a**) Protein expression; (**b**) Representative confocal microscopy images of GLUT4. Scale bar 75µm, Bule-DAPI, red-GLUT4. # *p* < 0.05, ## *p* < 0.01 vs. Normal group; * *p* < 0.05, ** *p* < 0.01 vs. Model group.

**Figure 4 molecules-28-00968-f004:**
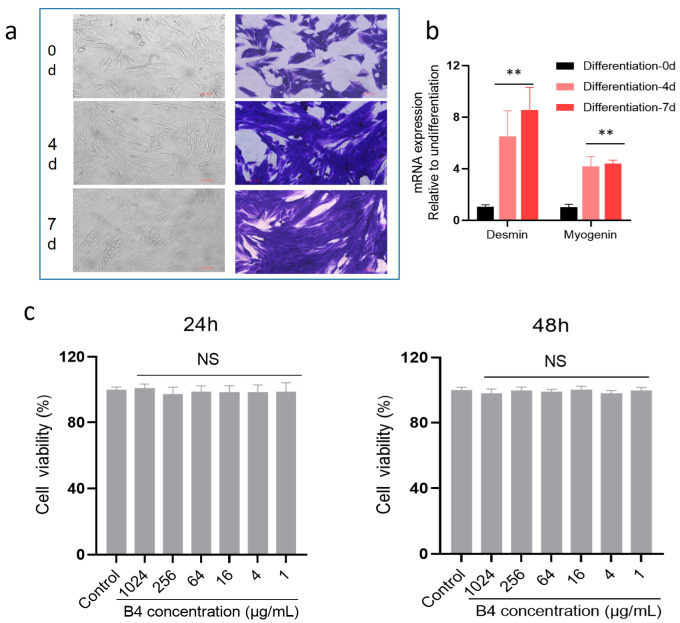
Differentiation of L6 myoblasts and cell viability. (**a**) Days 0, 4, and 7 of L6 differentiation and cells were stained by coomassie brilliant blue; (**b**) mRNA expression of desmin and myogenin was detected by PCR assay; (**c**) Cell viability of L6 cells was examined after 24 and 48 h of B4 treatment. ** *p* < 0.01 vs. Differentiation 0d group. NS (no significance) vs. control group. Data represent the mean ± standard error of three separate experiments.

**Figure 5 molecules-28-00968-f005:**
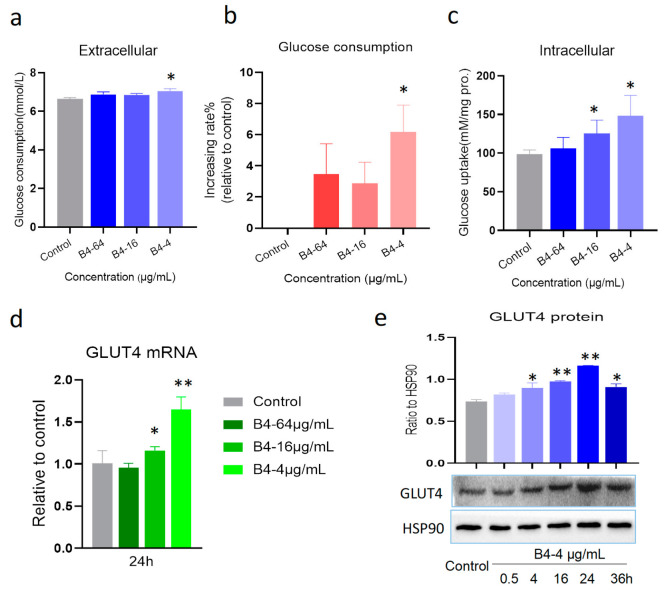
The glucose utilization and GLUT4 expression in L6 cells. (**a**) Glucose consumption; (**b**) Increasing rate of glucose consumption; (**c**) Glucose uptake; (**d**) mRNA expression of GLUT4; (**e**) Protein expression of GLUT4. * *p* < 0.05, ** *p* < 0.01 vs. normal group. Data represent the mean ± standard error of three separate experiments.

**Figure 6 molecules-28-00968-f006:**
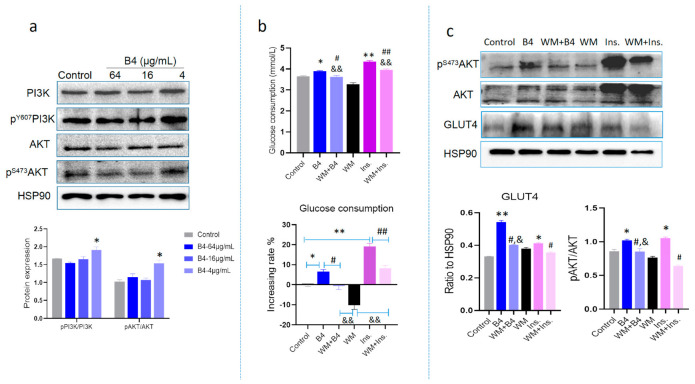
Protein expression and glucose consumption in the presence of B4 treated with the inhibitors or both in L6 cells. (**a**) The protein expression of PI3K/AKT in L6 cells; (**b**) Glucose consumption; (**c**) Western blot assay detected expression of the downstream protein of PI3K. * *p* < 0.05, ** *p* < 0.01 vs. control group; # *p* < 0.05, ## *p* < 0.01 vs. B4 or Ins. group; & *p* < 0.05, && *p* < 0.01 vs. WM group. Data represent the mean ± standard error of three separate experiments. Ins: inslin, WM: wortmannin.

**Figure 7 molecules-28-00968-f007:**
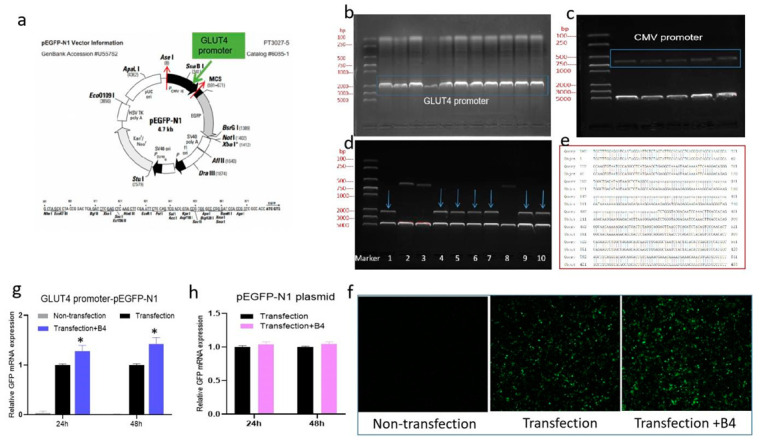
The effect of B4 on GLUT4 promoter plasmid. (**a**) The illustration of GLUT4 promoter replaced the CMV promoter of vector pEGFP-N1; (**b**) PCP amplification for GLUT4 promoter; (**c**) pEGFP-N1 for digestion; (**d**) Digestion of recombinant plasmids; (**e**) Gene sequencing and alignment; (**f**) Fluorescent intensity of GLUT4 promoter plasmid after B4 treatment for 48 h; (**g**) GFP mRNA expression of GLUT4 promoter plasmid; (**h**) GFP mRNA expression of pEGFP-N1 control plasmid after B4 treatment for 24 h and 48 h. * *p* < 0.05 vs. Transfection group.

**Figure 8 molecules-28-00968-f008:**
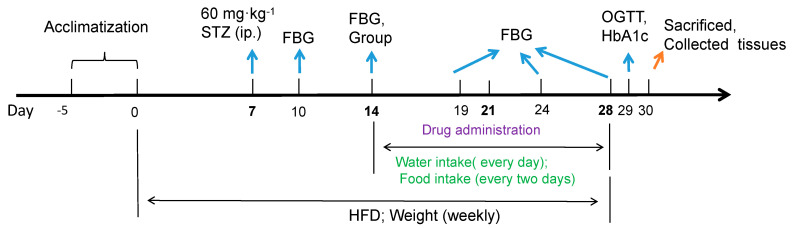
The experimental procedure. After 5 days of acclimatization in the Laboratory of Barrier Environment, SD rats were fed a high-fat diet, and one week later (day 7), 60 mg/kg STZ was injected intraperitoneally. The level of FBG was measured on the 3rd (day 10) and 7th (day 14) day after the STZ injection. The rats with the FBG ≧13.0 mmol/L were selected and divided into groups. The drug was given for two consecutive 2 weeks. During the administration, the water intake was measured every day, food intake was recorded every two days, and the body was weighed weekly. At the end of the administration, the OGTT was performed and the animals were sacrificed under anesthesia and the tissues were collected. STZ: Streptozotocin; HFD: high-fat diet; FBG: fasting blood glucose; OGTT: oral glucose tolerance test; ip.: intraperitoneal injection.

## Data Availability

The data presented in this study are available on request from the corresponding author.

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
