# Peer review of "Anemoside B4 Exerts Hypoglycemic Effect by Regulating the Expression of GLUT4 in HFD/STZ Rats"

_molecules, 2023, doi:10.3390/molecules28030968_

Round 1

Reviewer 1 Report

Gong et al., studied the effect of anemoside B4 on hyperglycemic rats and L6 myotubes. They found that B4 regulates phosphorylation of PI3K and Akt. Also, B4 promotes Glut4 transcription.

There are a few concerns regarding animal model and experimental design.

1.       Animal model: The authors have used HFD + Streptozotocin model for induction of diabetes. In this study, the HFD was given for one week. After one week of HFD feeding animals were injected with 60mg / kg body weight of streptozotocin. Kindly provide the reference for this method of induction. This dose of streptozotocin is more likely to induce pancreatic β-cell necrosis which may reflects type-1 diabetic model.

2.       If the authors are specifically interested in GLUT4 translocation, type-2 animal model could be more suitable one. Were the authors measured insulin levels in these animals? How was the animal’s response to insulin administration? Were these animals insulin resistant?

3.       Section 3.11: Glucose uptake assay: The authors mentioned that glucose was measured in cell supernatant (which is culture medium) as well as cell lysate. Fig 6a and Fig 6b show increased extracellular glucose levels than control [which indicates the uptake by cells may be lesser than control]. To contrast fig 6c shows intracellular glucose levels, which are higher than control. This is contradictory. Kindly clarify.

4.       Authors conclude that B4 could enhance Glut4 expression by activating PI3K / Akt signaling. PI3K/ Akt signaling mainly improves glut4 translocation rather than its transcription. The increased glut4 transcription is not equal to increased glucose uptake. The conclusion can be more clearer.  

5.       Spell check is required throughout the manuscript.

6.       Specify the phosphorylation sites of PI3K and Akt in western blot images.

7.       Section 3.1: Animals and cell lines: The authors mentioned human embryonic kidney cells 293T which was not used in this study.

Author Response

  1. Animal model: The authors have used HFD + Streptozotocin model for induction of diabetes. In this study, the HFD was given for one week. After one week of HFD feeding animals were injected with 60mg / kg body weight of streptozotocin. Kindly provide the reference for this method of induction. This dose of streptozotocin is more likely to induce pancreatic β-cell necrosis which may reflects type-1 diabetic model.
  2. If the authors are specifically interested in GLUT4 translocation, type-2 animal model could be more suitable one. Were the authors measured insulin levels in these animals? How was the animal’s response to insulin administration? Were these animals insulin resistant?

Reply (For Comments 1 and 2) : Diabetic model induced by STZ/HFD is closely related to the dosage of STZ and the duration of high-fat diet, as well as animal strain. Refer to the method mentioned in the literature, STZ dose of 150 mg/kg in mice and 45 mg/kg in rats was employed for the type 2 diabetes model, and recording to our previous experimental results, we concluded that STZ dose of 60 mg/kg combined with short-term high-fat diet (4 weeks) resulted in stable hyperglycemic rat models. ,We measured insulin levels at the end of the experiment, the results  showed that there was no significant difference in serum insulin level between model and normal rats (data was not shown), which was not consistent with the absolute deficiency of insulin of type 1 diabetes. In addition, the model rats responded to non-insulin therapy. These evidences showed that the hyperglycemic model in this paper is biased towards type 2 diabetes mellitus and may be advanced type 2 diabetes. The main object of this paper is to study the hypoglycemic effect and mechanism of B4, regardless of type 1 diabetes or type 2 diabetes, the main characteristic is hyperglycemia, which meets the research purpose of our study.

  1. Section 3.11: Glucose uptake assay: The authors mentioned that glucose was measured in cell supernatant (which is culture medium) as well as cell lysate. Fig 6a and Fig 6b show increased extracellular glucose levels than control [which indicates the uptake by cells may be lesser than control]. To contrast fig 6c shows intracellular glucose levels, which are higher than control. This is contradictory. Kindly clarify.

Reply: Figure 6a and 6b showed the glucose consumption (extracellular), calculated from the initial concentration of the culture medium minus the concentration of the cell supernatant. The consumption of glucose reflects the glucose uptake of the cell. Figure 6c showed the glucose content of the cell lysate (intracellular), which represents the glucose uptake. Increased glucose consumption means increased glucose uptake by cells.

  1. Authors conclude that B4 could enhance Glut4 expression by activating PI3K / Akt signaling. PI3K/ Akt signaling mainly improves glut4 translocation rather than its transcription. The increased glut4 transcription is not equal to increased glucose uptake. The conclusion can be more clearer.  

Reply: PI3K/AKT increases the amount of GLUT4 and promotes membrane translocation, allowing tissue glucose uptake. That is, this signaling pathway affects both the transport function and the expression of GLUT4. Studies have shown that overexpression of GLUT4 can promote glucose transport and decrease blood glucose in transgenic rats, which means that increased expression of GLUT4 will affect glucose transport function. When a gene changed at transcription and protein levels, its function is also affected. This study focused on GLUT4 expression to investigate the role of B4 in lowering blood glucose in hyperglycemic rats. We made some adjustments to the statement in the conclusion and the modifications are highlighted in red.

  1. Spell check is required throughout the manuscript.

Reply: Thank you for your careful review. We have checked the spelling of this manuscript, the modification trace is shown in the text.

  1. Specify the phosphorylation sites of PI3K and Akt in western blot images.

Reply: The phosphorylation site of PI3K and AKT has been marked in the WB images in Figure 4 and Figure 7.

  1. Section 3.1: Animals and cell lines: The authors mentioned human embryonic kidney cells 293T which was not used in this study.

Reply: Human embryonic kidney cells 293T was used in construction of GLUT4 promoter plasmid, described in method 3.13.

Reviewer 2 Report

The proposal is interesting, it describes a proposal for the mechanism by which anemoside B4 exerts a hypoglycemic effect. Participation of the increase in GLUT4 expression is shown by clear and concise experiments.

A change of title is suggested, focusing on the anemoside; a proposal is:

Anemoside B4 exerts hyperglycemic effects by regulating GLUT4 expression in HFD/STZ rats.

The introduction is adequate, and shows the previous studies carried out, which have a wide relationship with the study.

The results are adequately described; however it is necessary to indicate the differences between FBG and FBG in the OGTT. For example, in Figure 2 section b) the FBG quantification is mentioned and in Figure 2 section d) the FBG quantification to build an OGTT curve is mentioned, what is the difference?

This same figure (2) is recommended to represent the values of the OGTT in differences vs. initial time, this in order to be able to observe more clearly the hypoglycemic effect exerted by B4.

No mention is made of what is attributed no changes in the consumption of water and food have been observed. Taking into account the symptoms described in the animal model (polyphagia and polyuria), describe in the discussion section.

It is recommended to modify the description of the results relating to the evidence of GLUT4; mention the effects on the cell line, the effects on transcription and finally the effect on the muscle tissue of the animal model; in this way the results will be clearer.

Other saponins with hypoglycemic effects have been described in the literature, which are administered orally or intragastrically; Justification of the selected administration route and the use of dapaglifozin is requested. Include this information in the discussion section as it plays a relevant role in the effects on GLUT4 expression obtained.

Author Response

A change of title is suggested, focusing on the anemoside; a proposal is:

Anemoside B4 exerts hyperglycemic effects by regulating GLUT4 expression in HFD/STZ rats.

Reply: Thank you for your comments and we accepted your suggestion to amend the title to "Anemoside B4 exerts hypoglycemic effect by regulating the expression of GLUT4 in HFD/STZ rats".

The introduction is adequate, and shows the previous studies carried out, which have a wide relationship with the study.

 Reply: Thank you very much for your affirmation.

The results are adequately described; however it is necessary to indicate the differences between FBG and FBG in the OGTT. For example, in Figure 2 section b) the FBG quantification is mentioned and in Figure 2 section d) the FBG quantification to build an OGTT curve is mentioned, what is the difference?

Reply: FBG in Figure 2b and Figure 2d both represent fasting blood glucose values.  In Figure 2b, the blood collected from the rat tail tip and FBG was measured by a blood glucose meter. FBG in Figure 2d was measured by automatic biochemical analyzer, blood samples were taken from the rat ocular venous plexus for OGTT test.

This same figure (2) is recommended to represent the values of the OGTT in differences vs. initial time, this in order to be able to observe more clearly the hypoglycemic effect exerted by B4.

Reply: In Figure 2d, the time on the abscissa corresponds to the blood glucose level at different time points within 0-120min of oral glucose in the OGTT test (OGTT was performed at day 29 ,please see schedule in Figure 1) . In order to observed more clearly the hypoglycemic time of B4, the time of figure title in Figure 2b has been modified to be consistent with that in Figure 2a.

No mention is made of what is attributed no changes in the consumption of water and food have been observed. Taking into account the symptoms described in the animal model (polyphagia and polyuria), describe in the discussion section.

Reply: Relevant contents have been described in the second paragraph of discussion section, and remarked in red.

It is recommended to modify the description of the results relating to the evidence of GLUT4; mention the effects on the cell line, the effects on transcription and finally the effect on the muscle tissue of the animal model; in this way the results will be clearer.

Reply: Thank you for your suggestion. After careful consideration, we believe that the order of description of the results related to GLUT4 was appropriate. We consider that, first, the up-regulation of GLUT4 and its upstream signaling pathway PI3K/AKT was discovered in animal experiments. Then, the same phenomenon was observed at the cellular level, and the regulatory mechanism of the up-regulation of GLUT4 expression was investigated. Through inhibitor and promoter plasmid construction experiments, we found that the enhancement of GLUT4 was not only regulated by the PI3K/AKT pathway, but also related to the enhanced transcriptional activity of GLUT4.

Other saponins with hypoglycemic effects have been described in the literature, which are administered orally or intragastrically; Justification of the selected administration route and the use of dapaglifozin is requested. Include this information in the discussion section as it plays a relevant role in the effects on GLUT4 expression obtained.

Reply: The description of administration route of B4 and the use of dapaglifozin had been added in the second paragraph of discussion section and marked in red.

Round 2

Reviewer 1 Report

The authors have made necessary changes.

Author Response

I truly appriciate your comments and suggestions!